



# Cloud_cci ATSR-2 and AATSR dataset version 3: a 17-year climatology of global cloud and radiation properties

Caroline A. Poulsen[1], Gregory R. McGarragh[2], Gareth E. Thomas[3,6], Martin Stengel[4], Matthew W. Christensen[5], Adam C. Povey[7], Simon R. Proud[5,6], Elisa Carboni[3,6], Rainer Hollmann[4], and Roy G. Grainger[7]

[1]Monash University, Melbourne, Australia
[2]Department of Physics, University of Oxford, Clarendon Laboratory, Parks Road, Oxford OX1 3PU, UK Now at Cooperative Institute for Research in the Atmosphere, Colorado State University
[3]Science Technology Facility Council, Rutherford Appleton Laboratory, Harwell, Oxfordshire, UK
[4]Deutscher Wetterdienst, Frankfurter Str. 135, 63067, Offenbach, Germany
[5]Department of Physics, University of Oxford, Clarendon Laboratory, Parks Road, Oxford OX1 3PU, UK
[6]National Centre for Earth Observation, Reading RG6 6BB, UK
[7]National Centre for Earth Observation, Atmospheric, Oceanic and Planetary Physics, University of Oxford, Parks Road, Oxford OX1 3PU, U.K.

**Correspondence:** Caroline Poulsen (caroline.poulsen@monash.edu)

**Abstract.** We present version 3 (V3) of the Cloud_cci ATSR-2/AATSR dataset. The dataset was created for the European Space Agency (ESA) Cloud_cci (Climate Change Initiative) program. The cloud properties were retrieved from the second Along-Track Scanning Radiometer (ATSR-2) on board the second European Remote Sensing Satellite (ERS-2) spanning 1995-2003 and the Advanced ATSR (AATSR) on board Envisat, which spanned 2002-2012. The data comprises a comprehensive set

of cloud properties: cloud top height, temperature, pressure, spectral albedo, cloud effective emissivity, effective radius and optical thickness alongside derived liquid and ice water path. Each retrieval is provided with its associated uncertainty. The cloud property retrievals are accompanied by high-resolution top and bottom-of-atmosphere short- and long-wave fluxes that have been derived from the retrieved cloud properties using a radiative transfer model. The fluxes were generated for all-sky and clear-sky conditions. V3 differs from the previous version 2 (V2) through development of the retrieval algorithm and

attention to the consistency between the ATSR-2 and AATSR instruments. The cloud properties show improved accuracy in validation and better consistency between the two instruments, as demonstrated by a comparison of cloud mask and cloud height with collocated CALIPSO data. The cloud masking has improved significantly, particularly the ability to detect clear pixels The Kuiper Skill score has increased from .49 to .66. The cloud top height accuracy is relatively unchanged. The AATSR liquid water path was compared with the Multisensor Advanced Climatology of Liquid Water Path (MAC-LWP) in

regions of stratocumulous cloud and shown to have very good agreement and improved consistency between ATSR-2 and AATSR instruments, the Correlation with MAC-LWP increase from .4 to over .8 for these cloud regions. The flux products are compared with NASA Clouds and the Earth's Radiant Energy System (CERES) data, showing good agreement within the uncertainty. The new dataset is well suited to a wide range of climate applications, such as comparison with climate models, investigation of trends in cloud properties, understanding aerosol-cloud interactions, and providing contextual information for



collocated ATSR-2/AATSR surface temperature and aerosol products. For the Cloud_cci ATSR-2/AATSR dataset a new digital identifier has been issued: https://doi.org/10.5676/DWD/ESA_Cloud_cci/ATSR2-AATSR/V003  Poulsen et al. (2019).

## 1   Introduction

Clouds play a critical role in the Earth's radiation budget as their response to the changing climate can cool or warm the planet. There is considerable uncertainty in the balance between these cooling and warming effects. The Fifth Assessment Report

of the Intergovernmental Panel summarised the current understanding of climate sensitivity, which measures the temperature change when the amount of carbon dioxide ($CO_2$) in the atmosphere is doubled. IPCC (2013) estimated this number to be between 1.5 and 4.5°C. The large range results almost entirely from the response of clouds. In terms of radiative impact the effect of cloud-aerosol interactions is also a major uncertainty. It is imperative to create accurate records of cloud properties and use them to study changes in cloud behaviour.

A number of satellite cloud records exist to address this question. The longest series of satellite instruments used to measure cloud comes from the Advanced Very High Resolution Radiometers (AVHRRs). Satellite cloud climatologies based on these instruments include: the International Satellite Cloud Climatology Project (ISCCP; Young et al., 2018), which also includes geostationary satellites; the AVHRR PATMOS-X climatology (Heidinger et al., 2014); the European Meteorological Satellite Organisation (EUMETSAT) Climate Monitoring Satellite Applications Facility (CM-SAF) CLARA-A2 dataset (Karlsson et

al., 2017); and the Cloud_cci AVHRR dataset (Stengel et al., 2019). Much attention has been focused on improving the quality of the Fundamental Climate Data Record (FCDR), i.e. the radiances, harmonising the calibration of instruments on different platforms and accounting for the impact of the diurnal cycle and drifting orbits. Algorithms are increasingly complex and more accurate. Nevertheless, there are significant differences between the products and their associated trends, as has recently been shown for the cloud mask in a study comparing the cloud fraction in four of the longest AVHRR datasets (Karlsson

and Devasthale, 2018). The Moderate Resolution Imaging SpectroRadiometer (MODIS) cloud record (Platnick et al., 2017; Baum et al., 2012) has much higher quality radiances but a shorter record beginning in 2002 and, similar to the Multi-angle Imaging SpectroRadiometer (MISR) dataset (Davies et al., 2017), considerable uncertainty (Marchand, 2013). Since 2006, CloudSat (Stephens et al., 2008) and CALIPSO (Winker et al., 2009), an active radar and lidar respectively, have collected vertical profiles of cloud. These have been of immense value in understanding clouds and climate processes, but their coverage

is sparse compared to a passive instrument and they have operated for a relatively brief period. It has been estimated that CloudSat-like radar instruments would need to constantly observe the Earth until at least 2030 to detect a noticeable trend in cloud top height related to climate change (Takahashi et al., 2019).

    Spanning a gap in time between AVHRR and MODIS, the ATSR-2/AATSR instrument series has the potential to offer a much more stable cloud property record than AVHRR. The ATSR-2/AATSR are part of a well characterised series of instru-

ments, using on-board calibration and posthoc vicarious calibration activities. These instruments' orbits are very similar and stable (see Table 1), which is key in climate applications. While the AATSR instrument ceased operation in 2012, the next instrument in the series, the Sea and Land Surface Temperature Radiometer (SLSTR), has been in operation since 2016 with



**Table 1.** Outline of the key specifications of the ATSR-2 and AATSR instruments compared to the follow-on SLSTR instruments. LTND is Local Time Descending Node. [1] 300 km swath over sea, 512 over land. [2] 1420 km nadir swath and 750 backwards (dual) view swath. [3] 0.5 km visible channel resolution and 1 km infrared channel resolution.

| Instrument specifications | | | | | |
|---|---|---|---|---|---|
| Instrument/Platform | LTDN | Swath (km) | Resolution (km) | Start | End |
| ATSR-2 ERS-2 | 10.30 | 300/512[1] | 1 | 06/1995 | 08/2008 |
| AATSR Envisat | 10.00 | 512/512 | 1 | 03/2002 | 04/2012 |
| SLSTR Sentinel-3a | 10.00 | 750/1420[2] | 0.5–1[3] | 02/2016 | |
| SLSTR Sentinel-3b | 10.00 | 750/1420[2] | 0.5–1[3] | 04/2018 | |

a second instrument launched in 2018. The instrument will continue for the foreseeable future as an ESA operational mission on Sentinel-3 platforms (Coppo et al., 2010). These satellite records have been used to produce the first climatology of top-

and bottom-of-atmosphere radiative flux collocated with the cloud products. This was derived from the Aerosol_cci (Thomas et al., 2009) and the Cloud_cci products combined with MODIS surface albedo and temperature profiles from ERA-Interim reanalysis which was then input into a radiative flux model. This climatology is produced at pixel resolution, i.e. $\sim 1$ km, which is high resolution compared to fluxes from the Clouds and the Earth's Radiant Energy System (CERES).

This paper documents production of the ATSR-2/AATSR cloud and flux property dataset, completed as part of the ESA

Cloud_cci program (Hollmann et al., 2013). The dataset is named ATSR-2/AATSRv3 and follows on from the precursor dataset ATSR-2/AATSRv2. It covers a 17-year time period from 1995-2012 and delivers cloud properties of superior quality to the previous version and additional flux products. The dataset has already been used in a number of studies, such as Neubauer et al. (2017), Christensen et al. (2017) and Zelinka et al. (2018). The following sections describe updates to the cloud algorithm and briefly introduce the products and their validation.

## 65  2  ATSR-2/AATSR instruments

The ATSR series of instruments are a multi-channel (0.55, 0.66, 0.87, 1.6, 3.7, 11 and 12 µm), dual-view imaging radiometer with the principal objective of measuring global surface temperature, aerosols and clouds. ATSR-2 was launched on-board ESA's ERS-2 spacecraft on 21 April 1995, while AATSR was launched on-board Envisat in 2002, with final measurements on 9 May 2012. A follow-on instrument, SLSTR, was launched in February 2016 on-board Sentinel-3A and a companion

instrument was launched in April 2018 on-board Sentinel-3B. The primary operational products are aerosol, land and sea surface temperature.

ATSR is designed to be low noise and measurements are carried out with a high level of accuracy as they include an on-board thermal black body and a visible calibration system designed for high uniformity and stability. The on-board calibration is supplemented by vicarious calibration with ground targets (Smith and Cox, 2013). A high level of stability is maintained in

the satellite's orbits through regular orbit control manoeuvres.



## 3 Cloud products

The same cloud variables are produced in V3 as in V2, but the flux products are new for V3. The variables, naming abbreviation, units, and algorithm type are summarised in Table 2. The data is available on three processing levels:

- **Level-2**: Retrieved cloud and flux variables at satellite sensor pixel level, being the same resolution and location as the sensor measurements (Level-1), i.e. approximately 1 km pixels.

- **Level-3U**: Cloud and flux properties of Level-2 orbits projected onto a global spatial grid without combining any observations from overlapping orbits, only sub-sampling. These products use a latitude-longitude grid of 0.05° resolution.

- **Level-3C**: Cloud and flux properties of Level-2 orbits from a single sensor combined (averaged / sampled for histograms) onto a global spatial grid. The temporal resolution of this product is 1 month. These products use a latitude-longitude grid of 0.5° resolution.

In addition to cloud properties, each of the retrieved cloud variables includes pixel-level uncertainties. The propagation of those from Level-2 to 3 is described in Stengel et al. (2017).

### 3.1 Algorithm

The ATSR-2/AATSR cloud products were produced using the Community Cloud retrieval for Climate (CC4CL) algorithm, developed during the ESA CCI program. The algorithm has been described in detail in Stengel et al. (2017); Poulsen et al. (2012); McGarragh et al. (2017); Sus et al. (2018). For completeness, the basic concepts are summarised here. The CC4CL algorithm consists of three main components: (1) cloud detection, (2) cloud typing and (3) the retrieval of cloud properties based on an Optimal Estimation (OE) technique. The cloud mask and cloud phase are both determined using Artificial Neural Network (ANN) algorithms. Each ANN is trained using CALIPSO data collocated with AVHRR and then transferred to the ATSR series of instruments through the application of spectral band adjustments described in Sus et al. (2018).

The optimal estimation retrieval within CC4CL, known as the Optimal Retrieval of Aerosol and Cloud (ORAC), is a non-linear statistical inversion method based on Bayes' theorem (Rodgers, 2000). A state vector containing all variables to be retrieved is optimised to obtain the best fit between observed top-of-atmosphere (TOA) radiances and those simulated by a forward model. The inversion can accommodate a priori information and its associated uncertainty (though, in this application, only surface temperature is constrained, based on ERA-Interim reanalyses). The method provides a rigorous characterisation of the retrieval uncertainties, including propagation of measurement noise, the uncertainty in parameters assumed by the model and the uncertainty in the forward model itself. The retrieval also provides information about the quality of the fitting, such as the number of iterations it took to minimise the retrieval to an acceptable level and cost. Similar to a $\chi^2$ statistic, cost is a combination of the squared deviations between the measurements and forward model as well as the retrieved state vector and a priori state vector, each weighted by their associated covariance matrix. The cost provides an indication of how well the measurements fit the model. A cost close to one indicates the results have fit the model well, a high cost e.g. greater than 10 indicates a poor fit of the measurements to the model.





**Table 2.** Cloud_cci ATSR-2/AATSR cloud and radiation properties. $\text{ANN}_{\text{mask}}$ = Artificial Neural Network for cloud detection, $\text{ANN}_{\text{phase}}$ = Artificial Neural Network for cloud phase, SV = state vector, PP = post processed, OE = optimal estimation, BR = BUGSRad (a radiative flux algorithm), TOA = top-of-atmosphere, BOA = bottom-of-atmosphere (i.e. the surface), LW = longwave, SW = shortwave. The upper part of the table has been adopted from Sus et al. (2018).

| Cloud properties | | | | |
|---|---|---|---|---|
| Variable name | Abbreviation | Units | Origin | Comment |
| Cloud mask fraction | CFC | 1 | $\text{ANN}_{\text{mask}}$ | Binary cloud classification |
| Cloud phase | CPH | 1 | $\text{ANN}_{\text{phase}}$ | Cloud phase classification |
| Cloud top pressure | CTP | hPa | SV | OE retrieval |
| Cloud top height | CTH | km | PP | Derived from CTP and atmospheric profile |
| Cloud top temperature | CTT | K | PP | Derived from CTP and atmospheric profile |
| Cloud effective radius | CER | µm | SV | OE retrieval |
| Cloud optical thickness | COT | 1 | SV | OE retrieval |
| Surface temperature | STEMP | K | SV | OE retrieval |
| Cloud water path | CWP | $\text{g m}^{-2}$ | PP | Derived from CER and COT (Stephens, 1978) |
| Cloud albedo at 0.6 µm | CLA | 1 | PP | Derived from CER and COT |
| Cloud effective emissivity | CEE | 1 | PP | Derived from CER and COT |
| Broadband flux properties | | | | |
| TOA up-welling SW flux | $\text{SWF}_{\text{TOA}}^{\text{up}}$, clear $\text{SWF}_{\text{TOA}}^{\text{up}}$ | $\text{W m}^{-2}$ | BR | All sky and clear sky |
| TOA up-welling LW flux | $\text{LWF}_{\text{TOA}}^{\text{up}}$, clear $\text{LWF}_{\text{TOA}}^{\text{up}}$ | $\text{W m}^{-2}$ | BR | All sky and clear sky |
| BOA up-welling SW flux | $\text{SWF}_{\text{BOA}}^{\text{up}}$, clear $\text{SWF}_{\text{BOA}}^{\text{up}}$ | $\text{W m}^{-2}$ | BR | All sky and clear sky |
| BOA up-welling LW flux | $\text{LWF}_{\text{BOA}}^{\text{up}}$, clear $\text{LWF}_{\text{BOA}}^{\text{up}}$ | $\text{W m}^{-2}$ | BR | All sky and clear sky |
| BOA down-welling SW flux | $\text{SWF}_{\text{BOA}}^{\text{up}}$, clear $\text{SWF}_{\text{BOA}}^{\text{up}}$ | $\text{W m}^{-2}$ | BR | All sky and clear sky |
| BOA down-welling LW flux | $\text{LWF}_{\text{BOA}}^{\text{up}}$, clear $\text{LWF}_{\text{BOA}}^{\text{up}}$ | $\text{W m}^{-2}$ | BR | All sky and clear sky |



The radiation products are created using BUGSRad (Stephens et al., 1991) in a similar manner to Fu and Liou (1992). BUGSRad is a radiative transfer algorithm based on the two-stream approximation and correlated-$k$ distribution methods of atmospheric radiative transfer. It is applied to a single-column atmosphere for which the cloud and aerosol layers are assumed to be plane-parallel. Cloud and aerosol properties retrieved using CC4CL together with collocated visible and near-infrared surface albedo from MODIS (Schaaf et al., 2002) are ingested into BUGSRad to compute both shortwave and longwave radiative fluxes for the top- and bottom-of-atmosphere. Total solar irradiance is drawn from the Solar and Heliospheric Observatory (SOHO; Domingo et al., 1995). The algorithm uses 18 bands that span the electromagnetic spectrum to compute the broadband flux. In total, 6 bands are used for shortwave and 12 bands are used for longwave radiative flux calculations. To account for the low sampling frequency of the polar orbiting satellite and the dependence of the shortwave fluxes on viewing geometry, an angular dependence correction is applied to the shortwave radiation properties to make the L3C monthly products represent 24 hour averages. Further details are outlined in Stengel et al. (2019).

Since V2 was produced, a number of developments have been made to the algorithm, resulting in considerable improvement to the ATSR-2/AATSR records, these are summarised below. Figures 1 and 2 show global maps of yearly average cloud properties from 2008 for V3 Level-3C data compared to that from V2.

– The cloud mask was retrained using a larger dataset including 1 km CALIPSO data. This has reduced the number of clouds falsely detected over polar regions (sea, sea ice and land) and, most notably, increased the number of clouds detected in stratocumulous cloud banks.

– The cloud phase selection in V2 used a threshold scheme developed by Heidinger and Pavolonis (2009). This has been replaced with an ANN approach for V3 (Stengel et al., 2019). The change significantly increased the number of clouds in the liquid phase and LWP, particularly in the northern and southern storm track regions. This change also affects the retrieval of cloud top height, with an overall reduction in the height of the clouds. This is particularly evident in the tropics and the stratocumulus cloud banks, accompanied by an increase in LWP.

– The surface reflectance model was revised to correct a bug in the application of large solar zenith angles. This resulted, where applicable, in a significant decrease in the COT and CER to much more realistic values.

– The Look-Up-Tables (LUTs) are now based on Baum et al. (2014) ice optical properties instead of Baran et al. (2004). This resulted in significantly smaller ice CER and COT, with a corresponding reduction in IWP to more realistic values.

– In V2, maintaining consistency with the earlier sections of the AVHRR record required using lower resolution (and less accurate) auxiliary datasets for ice and snow, such as European Centre for Medium Range Forecasting (ECMWF) reanalysis, and inferior land sea masks. This resulted in poor results over mountainous and snow or ice covered regions. These auxiliary datasets were also not consistent with those used by the AATSR ORAC Aerosol_cci. In V3, we implemented the higher resolution National Snow and Ice data center (NISE) masks (Brodzik and Stewart, 2016), improving retrievals over snow and ice covered surfaces.





– In V2 there was a discontinuity between the ATSR-2 and AATSR cloud retrievals, particularly in cloud fraction, COT and CER. This discontinuity was caused by a number of factors:

- The use of the 3.7 μm channel to generate the dataset, which differs in dynamic range between ATSR-2 and AATSR.

- Differences between the two instruments in the availability of shortwave channels across the swath during the day.

– In order to create a record which minimised the inconsistency between ATSR-2 and AATSR (and the aerosol record),
in V3 cloud properties were retrieved using the 1.6 μm channel rather than the 3.7 μm channel and only retrieved for the narrow swath mode of ATSR-2 when all channels are present.

The key strengths of the Cloud_cci datasets have been retained in V3.

- The spectral consistency of derived parameters, which is achieved by an OE approach based on a physically consistent cloud model simultaneously fitting satellite observations from the visible to the mid-infrared.

– Uncertainty characterisation, which is inferred at pixel level from OE theory, that is physically consistent (1) with the uncertainties of the input data (e.g. measurements, a priori) and (2) among the retrieved variables. These pixel-level uncertainties are further propagated into the monthly products.

- Comprehensive assessment and documentation of the retrieval schemes and the derived cloud property datasets including the exploitation of applicability for evaluation of climate models and reanalyses.

**4   Validation and Comparison**

An evaluation of CC4CL cloud mask and cloud top height Level-2 products has been carried out based on CALIPSO data for five days, covering all seasons, in 2008: 20 March, 13 June, 20 June, 21 September and 20 December. The cloud fraction and height validation was based on CALIPSO cloud observations which were simultaneously (i.e. within 5 minutes) observing the same location as the AATSR satellite. For morning satellites, such collocations only occur at high latitudes, i.e. greater that 70
degrees, restricting the comparison to the types of cloud found at those latitudes. Clouds in that latitude band are often located over snow and sea ice, which is a more difficult retrieval as both clouds and the surface are bright in the visible channels and cold in the infrared. Hence, these results should be considered a conservative validation, as the results for other regions, such as mid latitudes and tropics, particularly over ocean, are likely to be more accurate. The CALIPSO cloud products used in the validation study were the 1 km layer and 5 km layer products version 4-20.

**4.1   Cloud fraction**

The AATSR Level-2 cloud fraction products are retrieved at 1 km resolution so the retrievals were collocated with the CALIPSO 1 km cloud products. These are less sensitive to thin clouds than the 5 km products (CALIOP Quality statement, 2019), which were used in the evaluation of the Cloud_cci AVHRR products (Stengel et al., 2019). The validation was repeated using the



**Figure 1.** Examples from 2008 of Level-3C (yearly average) Cloud_cci AATSR V3 (left) and V2 (right). From the top: cloud fraction (CFC), liquid cloud faction (CPH), cloud optical thickness (COT) and cloud effective radius (CER),

5 km products (not shown) and the changes were negligible. The Kuiper skill score, a measure of correct cloud identification, is consistent with the results found for the AVHRR Cloud_cci product in the same region.

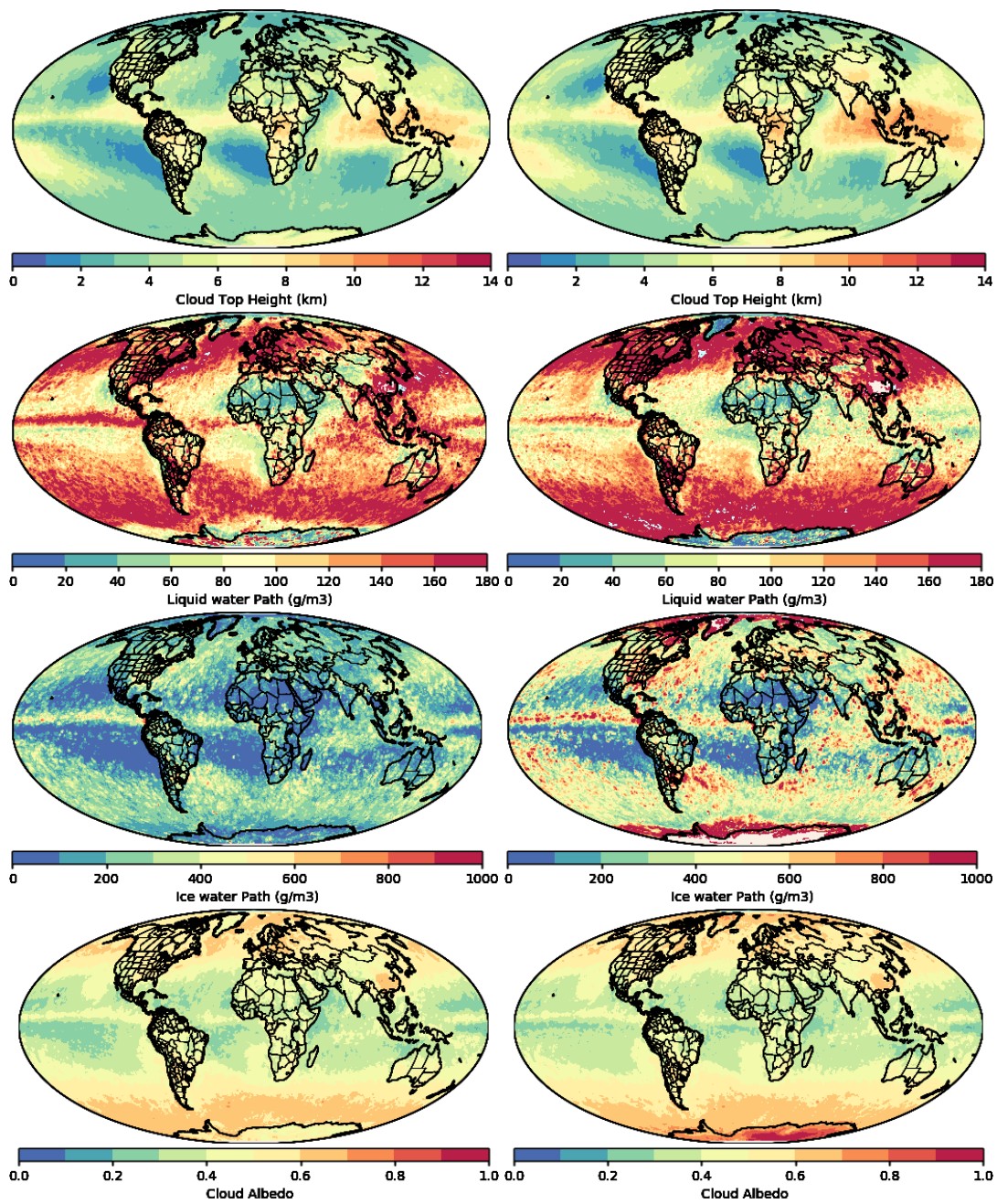

**Figure 2.** As Fig. 1 but for cloud top height (CTH), liquid water path (LWP), ice water path (IWP) and cloud albedo (CLA).

The results of the comparison are shown in Table 3. The cloud detection has clearly improved for V3, with the most significant improvement being the identification of clear pixels. This result is consistent with the reduction in cloud fraction observed over the poles in the global maps. The cloud detection improved for both clear and cloudy pixels. The identification of clear





pixels increased significantly from a probability of detection (POD) of 61% to 76%.

**Table 3.** Comparison of collocated AATSR cloud mask and CALIPSO 1 km layer product for V2 (left) and V3 (right). The comparison metrics shown are hit rate, Probability Of Detection (POD) for cloudy and clear pixels, and the Kuiper skill score (KSS) as well as the total number (cloudy and clear) of collocations used in the analysis

| AATSR Cloud fraction validation | | |
|---|---|---|
| Score | V2 | V3 |
| KSS | .49 | .66 |
| Hit rate | 79.9 | 86.0 |
| $POD_{cloud}$ | 88.1 | 90.4 |
| $POD_{clear}$ | 61.2 | 75.9 |
| Number | 23468 | 23468 |

### 4.2 Cloud top height

The Cloud top height product was validated using the CALIPSO 1 km product. In previous studies (Sus et al., 2018) it was shown that the CTH retrieval is more accurate when the cloud is opaque or single layer. Here, the opacity flag from the CALIPSO 1 km layer product is used to verify this finding. The opacity flag indicates features that completely attenuate the lidar beam (CALIOP Quality statement, 2019). Results are summarised in Table 4 and are presented separately for all cloud observations, only opaque clouds, the cloud top height corrected for penetration depth and for all clouds retrievals with a cost less than 5 (as would be expected for single layer clouds). From V2 to V3, the correlation with the lidar measurements for all the collocated pixels was unchanged but the bias decreased from 1.3 km to 0.9 km. When only opaque cloud are considered, the correlation increases considerably but with a slight increase in bias. The corrected cloud top heights are produced by approximating the observed brightness temperature as emitted from one optical depth into the cloud and assuming the cloud is vertically homogeneous with a constant lapse rate. This product achieves a similar correlation to the all-cloud results but further reduces the bias, from 0.94 to 0.85 km for V3. For clouds with a cost less than 5, the correlation decreased and the bias also reduced. High costs are indicative of multi-layer cloud, such as thin cirrus over liquid cloud. These are typically retrieved as some weighted average of the two layers, returning an nonphysical value (Poulsen et al., 2012). Overall, V3 is a superior cloud top height (temperature and pressure) product. The comparison was also performed with the CALIPSO 5 km layer (not shown) and the sensitivity to optical depth investigated. The results showed negligible variation with optical depth threshold and with the 5 km product.

### 4.3 Liquid water path

The liquid water path of Cloud_cci datasets is evaluated against the Multisensor Advanced Climatology of Liquid Water Path (MAC-LWP) dataset (Elsaesser et al., 2017) over ocean. The MAC-LWP climatology is based on retrievals from multiple





**Table 4.** Comparison of AATSR cloud top height with collocated CALIPSO measurements for five days in 2008. On the left is shown V2 and on the right V3. The results are shown for all observations, only opaque clouds (as defined by CALIPSO), the corrected cloud top height product, and retrievals with a cost less than 5. All values are in km.

| AATSR CTH validation | | | |
|---|---|---|---|
| | Score | V2 | V3 |
| All observations | Correlation | 0.72 | 0.72 |
| | Bias | 0.73 | 1.1 |
| | Standard deviation | 2.3 | 2.5 |
| Opaque Clouds | Correlation | 0.88 | 0.91 |
| | Bias | -0.06 | 0.2 |
| | Standard deviation | 1.4 | 1.2 |
| Corrected CTH | Correlation | 0.73 | 0.73 |
| | Bias | -0.6 | 0.95 |
| | Standard deviation | 2.2 | 2.4 |
| Cost < 5 | Correlation | 0.77 | 0.75 |
| | Bias | 0.44 | 0.96 |
| | Standard deviation | 2.0 | 2.3 |

microwave radiometer instruments, including the SSM/I series (Special Sensor Microwave Imager), the Tropical Rainfall Measurement Mission Microwave Imager (TMI), and the Advanced Microwave Scanning Radiometer for EOS (AMSR-E). Microwave measurements of LWP are typically more accurate than visible imagers because microwave instruments are able to penetrate deep convective clouds and ice over water clouds while also measuring the LWP at lower altitudes, which is not
possible for passive imagers. Their disadvantage is the large footprint, up to 0.25 degree.

This evaluation focuses on regions where liquid clouds are dominant (i.e. fewer than $5\%$ ice clouds), specifically three stratocumulus regions: the oceanic area west of Africa at $10°$-$20°$S, $0°$-$10°$E (SAF), the area west of South America at $16°$-$26°$S, $76°$-$86°$W (SAM), and the area west of California at $20°$-$30°$N, $120°$-$130°$W (NAM), similar to the analyses in PVIR (2018). The MAC-LWP dataset was collocated with the Cloud_cci dataset on a monthly basis. No correction was made for the
diurnal cycle as it is assumed to be small in the selected regions. A time series of the comparison is shown in Figure 3 and summarised in Table 5. There was a significant improvement from V2 to V3, particularly in the consistency between the ATSR-2 and AATSR instruments. The V2 dataset showed a large offset between ATSR-2 and AATSR which has almost disappeared in V3. The correlation with MAC-LWP exhibited in V3 is extremely good, over 0.8 for all regions which is a significant improvement over V2 particularly for the region off the Californian coast. The associated bias and standard deviation are also
very low typically less than $5\%$ of the total liquid water path.



**Figure 3.** Comparison of ATSR-2/AATSR Cloud_cci LWP time series (coloured) with MAC-LWP (black) for three regions: west of California (NAM, top), west of Africa (SAF, middle), and west of South America (SAM, bottom). Version 3 (left) has reduced the discontinuity between sensors seen in version 2 (right).

**Table 5.** Multi-annual (2000-2012) liquid water path validation results for ATSR-2/AATSR when compared with MAC-LWP monthly data for three regions of predominantly stratocumulous cloud. The results for V2 (left) and V3 (right) are compared for correlation, bias and standard deviation.

| Liquid water path validation with MAC-LWP | | |
|---|---|---|
| Score | ATSR-2/AATSR V2 | ATSR-2/AATSR V3 |
| West of Africa | | |
| Std (g/m$^2$) | 4.90 | 3.10 |
| Bias (g/m$^2$) | -8.18 | -3.39 |
| Correlation | 0.67 | 0.86 |
| South America | | |
| Std (g/m$^2$) | 5.46 | 3.13 |
| Bias (g/m$^2$) | -11.17 | -3.17 |
| Correlation | 0.57 | 0.82 |
| West of California | | |
| Std (g/m$^2$) | 4.64 | 2.91 |
| Bias (g/m$^2$) | -0.60 | 1.82 |
| Correlation | 0.41 | 0.80 |

## 4.4 Comparison of radiative fluxes

Examples of the Cloud_cci AATSR flux products are shown in Figure 4. On the left are the multi-month mean for 2008 from AATSR. The AATSR products are compared with the Clouds and Earth Radiation Energy System (CERES) Energy Balanced and Filled (EBAF) Top of atmosphere (TOA) and Bottom of Atmosphere (BOA) fluxes Edition 4.1 (Loeb et al., 2018). They
show very similar global patterns. The highest TOA longwave cloudy fluxes are observed over warm land, typically desert regions and ocean regions with low cloud coverage. The highest TOA longwave upwelling clear-sky fluxes are found in the tropics and mid-latitudes. The highest TOA cloudy shortwave fluxes are located over the bright polar regions, deserts and regions of bright cloud, such as the storm tracks and stratocumulus cloud decks. The TOA clear-sky shortwave fluxes are low over the oceans and higher over bright land surfaces. The global mean comparison of TOA fluxes is summarised in Table 6.
The means were compared for both $\pm 60°$ latitude and the whole globe, with the largest differences seen when the polar regions were included. The CERES data quality summary states that their all-sky shortwave and longwave monthly uncertainty is 2.5 and 2.5 $W\,m^{-2}$, respectively, while the clear-sky shortwave and longwave uncertainty is 5.4 and 4.6 $W\,m^{-2}$, respectively. The agreement between AATSR derived fluxes and CERES is within this uncertainty. AATSR all-sky longwave fluxes are slightly lower (colder) over the sea (particularly in the tropics, see red areas in the difference map) and slightly higher over
land. The TOA allsky shortwave flux shows the most differences with differences associated with different cloud regimes. The TOA shortwave flux in clear scenes is systematically lower than CERES indicating a potential underestimate of the surface





**Table 6.** Multi-annual (2003-2012), zonal averaged broadband shortwave and longwave fluxes (SWF, LWF) at the top-of-atmosphere (TOA) inferred from the Cloud_cci AATSR V3 dataset. Two latitude ranges, -60° to 60° (top) and -90° to 90° (bottom), are presented. The values are compared with the equivalent values from the Clouds and Earth Radiation Energy System (CERES) Energy Balanced and Filled (EBAF) fluxes. All values are given in $Wm^{-2}$. The differences and relative differences are also reported.

| TOA flux comparison with CERES | | | | |
|---|---|---|---|---|
| | $LWF_{TOA}^{up}$ | $SWF_{TOA}^{up}$ | Clear $LWF_{TOA}^{up}$ | Clear $SWF_{TOA}^{up}$ |
| Cloud_cci ATSR-2/AATSRv3 | 246.3 | 104.4 | 268.7 | 61.9 |
| CERES EBAF Ed 4.1 | 244.2 | 98.70 | 273.9 | 63.6 |
| Difference | -1.9 | -5.7 | 5.2 | 1.63 |
| Rel. difference | 0.8% | 5.7% | 1.9% | 2.6% |
| Cloud_cci ATSR-2/AATSRv3 | 234.9 | 114.0 | 255.1 | 47.5 |
| CERES EBAF Ed 4.1 | 225.1 | 104.2 | 248.9 | 48.7 |
| Difference | -9.9 | -9.8 | -6.2 | 1.2 |
| Rel. difference | 4.4% | 9.4% | 2.4% | 2.7% |

reflectance which uses a Cox and Munk (Cox and Munk , 1954) formulation, this will be investigated for future versions. The short wave and will be more sensitive to the diurnal cycle and illumination conditions.

The BOA longwave downwelling fluxes (all-sky and clear) have a minimum in the cold polar regions and a maximum in the
tropics. The corresponding shortwave fluxes are lowest in the southern and northern storm tracks and peak in the tropics. The BOA all-sky shortwave downwelling flux shows the largest regional differences. The BOA shortwave downwelling clear-sky fluxes show AATSR to be higher in regions of high aerosol loading. The global mean comparisons are summarised in Table 7. The means were compared for ±60° latitude and the whole globe, again finding the largest differences when the polar regions were included. The CERES data quality statement indicates uncertainties of 9 and 8 $W\,m^{-2}$ for longwave downwelling all-sky
and clear sky, respectively, and 14 and 6 $W\,m^{-2}$ for the shortwave BOA downwelling all-sky and clear sky, respectively. This is consistent with the AATSR values, although the difference between the AATSR and CERES products is considerably higher for shortwave observations in the polar regions. There, the diurnal correction is much more difficult to apply and the surface albedo used in the radiative transfer calculations is much more uncertain. While the AATSR fluxes also use satellite aerosol measurements in the clear sky calculations, the impact on the shortwave flux is less pronounced than in the CERES product,
which used MODIS aerosol products. The shortwave downwelling fluxes have the largest differences, but it is unclear which data set is more uncertain.





**Figure 4.** Examples of Level-3C (yearly average for 2008) Cloud_cci AATSR V3 (left column), CERES (middle column) and difference CERES-AATSR (right column) global maps of forcings from top to bottom $\mathrm{LWF}^{\mathrm{up}}_{\mathrm{TOA}}$, $\mathrm{LWF}^{\mathrm{up}}_{\mathrm{TOA}}$clear, $\mathrm{SWF}^{\mathrm{up}}_{\mathrm{TOA}}$ and $\mathrm{SWF}^{\mathrm{up}}_{\mathrm{TOA}}$clear

## 5 Conclusion

The AATSR-2/AATSR cloud data sets provide a unique data set that straddles the AVHRR and MODIS timelines and maintains
a stable orbit between satellite platforms. Version 3 of the Cloud_cci ATSR-2/AATSR cloud and radiation property dataset,
as presented in this paper, includes a number of algorithm improvements and bug fixes that positively impact the accuracy of
the cloud properties and improve the consistency between the ATSR-2 and AATSR instruments. The radiation properties are
new for version 3. The simultaneous provision of cloud, aerosol and radiative fluxes facilitates understanding the changes in
radiative flux associated with cloud properties at high resolution (1 km).



**Figure 5.** Examples of Level-3C (yearly average for 2008) Cloud_cci AATSR V3 (left column) and CERES (middle column) and difference CERES-AATSR in the right column global maps of forcing from the top to the bottom, $\text{LWF}_{\text{BOA}}^{\text{down}}$, $\text{LWF}_{\text{BOA}}^{\text{down}}$clear, $\text{SWF}_{\text{BOA}}^{\text{down}}$, $\text{SWF}_{\text{BOA}}^{\text{down}}$clear

Cloud fraction and cloud top height have been validated using CALIPSO measurements. While the lidar only finds good

collocations in the polar regions, the comparison demonstrates some of the key changes between V2 to V3. The cloud fraction shows considerable improvement in its ability to discern clear scenes, with the Kuiper Skill Score improving from 0.49 to 0.66. There were no major developments from V2 to V3 that would significantly affect the cloud top height retrievals, so the cloud top height validation has remained similar.

The MAC-LWP product has been compared with the ATSR-2 and AATSR product in regions of stratocumulous cloud.

The V3 data set shows significantly improved consistency between ATSR-2 and AATSR resulting from changes in the chan-





**Table 7.** As for Table. 6 but for the bottom-of-atmosphere (BOA).

| BOA flux comparison with CERES | | | | |
|---|---|---|---|---|
| | LWF$_{BOA}^{down}$ | SWF$_{BOA}^{down}$ | clearLWF$_{BOA}^{down}$ | clearSWF$_{BOA}^{down}$ |
| Cloud_cci ATSR-2/AATSRv3 | 364.6 | 192.2 | 335.3 | 255.7 |
| CERES EBAF Ed 4.1 | 354.4 | 190.4 | 323.9 | 250.3 |
| Difference | -10.26 | 1.9 | -11.4 | -5.4 |
| Rel. Difference | 2.9% | .97% | 3.5% | 2.1% |
| Cloud_cci ATSR-2/AATSRv3 | 334.1 | 181.3 | 301.7 | 241.3 |
| CERES EBAF Ed 4.1 | 306.8 | 176.0 | 272.7 | 232.6 |
| Difference | -28.4 | -5.3 | -29.1 | -8.7 |
| Rel. Difference | 9.2% | 3.0% | 10% | 3.8% |

nel selection. The ATSR-2/AATSR liquid water path is shown to be highly correlated with the MAC-LWP in these regions (coefficients > 0.8). The bias and standard deviation have reduced by around 5–10% in all regions.

The TOA and BOA flux products have been compared with the latest CERES EBAF version 4.1 products and show good agreement, within the estimated uncertainties. Differences are largest, and the most uncertain, over polar regions.

## 6 Data availability

A DOI has been issued for the dataset Cloud_cci AATSR/ATSR-2v3 described in this paper: https://doi.org/10.5676/DWD/ ESA_Cloud_cci/ATSR2-AATSR/V003  Poulsen et al. (2019). The CC4CL retrieval system used to produce the data version controlled and accessible at https://github.com/ORAC-CC/orac/wiki.

*Author contributions.* CP coordinated the generation of the presented dataset, which was undertaken by GT and EC, contributed to key
developments of the algorithm, evaluated the data and drafted the manuscript. MS developed the cloud detection and phase determination. GM contributed key developments to the algorithm, AP, SP, RG, GT, contributed to the development of the optimal estimation scheme. MC developed the radiation scheme. All authors assisted in drafting the manuscript.

*Competing interests.* The authors declare that no competing interests are present.

*Acknowledgements.* This work was undertaken as part of the ESA CCI program. The Natural Environment Research Council (NERC)
provides national capability funding for the National Centre for Earth Observation (NCEO) funding through contract number PR140015.



The data was generated and archived at the Centre for Environmental Data Analysis (CEDA), a which is also supported by NERC. The MAC-LWP and CERES data were obtained from the NASA Langley Research Center Atmospheric Science Data Center.



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
