# Peer review of "Cloud\_cci ATSR-2 and AATSR dataset version 3: a 17-year climatology of global cloud and radiation properties"

_Earth System Science Data, 2019_

## Referee Comment (RC1) · Anonymous Referee #1 · 6 Jan 2020

This manuscript documents the introduction to the version 3 (V3) of the Cloud_cci ATSR-2/AATSR dataset. Clouds data products are important for clouds-related climate and weather studies. Obtaining clouds parameters in modeling of climate is critical for predicting the temperature trend of the atmosphere and earth surface. Thus, this document and the data the manuscript wants to report, are important data for climate studies. This paper gives sufficient review of the historical instruments and products for this issue, English and presentation are both good. As a introduction to a dataset, it aslo outlines a general picture of the shape of the data. However, this paper's V3 data do not show very significant improvements from V2 as displayed by Figs1-2 and Table 4. So this manuscript may consider changing the title to address the difference

of V3 and V2. Also, the flux data of V3 have very big difference from CERES data as shown in Tables 6 and 7, up to ∼10%. But the authors claim their product agrees well with CERES. This must be changed. Also, why V3 flux and CERES flux have so big difference, must be unambiguously clarified. V3 or CERES problem? If flux data have 10% error, the data generally have no use for climate studies. Therefore, this reviewer recommends this paper be published after major revisions.

—————————————————

---

## Referee Comment (RC2) · Anonymous Referee #2 · 17 Jan 2020

This paper describes a new version of a cloud and radiative flux data set. Overall, I don't see any major problems with the paper and judge it appropriate for publication after consideration of the minor comments I list below.

In Figures 1 and 2, I recommend adding a third column which plots differences between the versions.

Table 3 contains a bunch of metrics that I don't know what they are. I think it would be good to give a one-sentence definition of KSS, hit rate, and POD.

Line 181 (and rest of paragraph): I don't know what "cost" is referring to, so "cost less than 5" makes no sense.

Line 188: they state here that thin multi-layer clouds are retrieved as the weighted average of the two layers. It would be useful to say if this situation is flagged in the data or if there's some mechanism to screen the data for this.

From line 221 to the end of the section: they discuss hear the uncertainty of the radiative flux estimates. Does uncertainty here refer to precision or accuracy? Also, are the values given for CERES (5.4 and 4.6 W/m2) the uncertainty for the global average or at a grid point?

Captions of Fig. 4 and 5. The captions refer to "forcing", but they really mean are fluxes. This should be changed.

---

## Referee Comment (RC3) · Anonymous Referee #3 · 20 Jan 2020

This paper describes Version 3 of a dataset from the Cloud-cci project based on ATSR-2/AATSR observations. The article is well structured and clear. Sufficient figures are included to understand the nature of the product and changes since Version 2. Multi-year global cloud data sets such as this are important for a wide variety of studies. Over the time period the observations were made, the capabilities of these two sensors offer several unique strengths. Further, this 17-year data set will be extended by observations from the two SLSTR instruments which began in 2016. Thus, the ATSR/AATSR datasets which are the subject of this paper are unique and important.

The paper includes sufficient references to the instruments and algorithms. Datasets

such as these require instruments with radiometric calibrations which are consistent and stable. Long term calibration drifts must be monitored and characterized. One paper on solar calibration accuracy is referenced but references on calibration of the infrared radiances are necessary, particularly in light of the statement that the dataset is well suited to investigation of trends.

Specific comments: The sentence in lines 45-47 implies CloudSat detection of GHG-forced trends in cloud height is limited by nadir-only sampling. The larger issue is separating forced cloud changes from natural variability. There has been considerable work which makes clear that long-term calibration stability is a major difficulty in characterizing forced trends from passive sensors observations. See, for example, Shea et al. (JGR, 2017, doi:10.1175/JCLI-D-16-0429.1). I'm not sure what the point of the last half of this paragraph is, which raises complex observational issues which are beyond the scope of this paper.

Line 72: This doesn't look like a complete sentence. Maybe something like "ATSR is designed to provide low noise radiance measurements . . ." - ?

Table 3: Define 'hit rate'. How is hit rate different from probability of detection? I'm not familiar with Kuiper Skill Score, please provide a reference.

Line 221: As a general comment: peer reviewed publications should be cited rather than data quality summaries posted on-line, unless the information is only available from on-line summaries. Results reported in refereed papers are archival, have usually received more scrutiny, and tend to be better explained and documented.

Line 222: Are the CERES uncertainties which are mentioned the uncertainties in the monthly global means?

Line 223 and 258-259: global means are within CERES uncertainties only for 60N-60S. All-sky fluxes show differences which are much larger, and there are significant regional biases which seem to be associated with clouds.

[Figure]

Line 164: "version 4-20" should be "version 4.20"

Line 228: incomplete sentence
* * *

---

## Author Comment (AC1) · 12 Jun 2020

The authors thank the reviewer for the comments which have improved the paper. The reviewer's comments are shown in red and the authors response shown in black.

However, this paper's V3

data do not show very significant improvements from V2 as displayed by Figs1-2 and

Table 4.

We noted that another reviewer suggested to include difference plots for Figures 1 and 2. These have been included and do indeed show more clearly that there has been a significant change between versions, particularly the microphysical cloud properties.

New figure above, old figure below

[Figure]

Figure 1. Examples from 2008 of Level-3C (yearly average) Cloud_cci AATSR V3 (left), V2 (middle) and difference of V3-V2 (right). From the top: cloud fraction (CFC), liquid cloud faction (CPH), cloud optical thickness (COT) and cloud effective radius (CER),

[Figure]

Figure 2. As Fig. 1 but for cloud top height (CTH), liquid water path (LWP), ice water path (IWP) and cloud albedo (CLA).

[Figure]

Figure 1. Examples from 2008 of Level-3C (yearly average) Cloud_cci AATSR V3 (left) and V2 (right). From the top: cloud fraction (CFC), liquid cloud faction (CPH), cloud optical thickness (COT) and cloud effective radius (CER),

[Figure]

Figure 2. As Fig. 1 but for cloud top height (CTH), liquid water path (LWP), ice water path (IWP) and cloud albedo (CLA).

Also, the flux data of V3 have very big difference from CERES data as

shown in Tables 6 and 7, up to ~10%. But the authors claim their product agrees well

with CERES. This must be changed.

Also, why V3 flux and CERES flux have so big

difference, must be unambiguously clarified. V3 or CERES problem? If flux data have

10% error, the data generally have no use for climate studies. Therefore, this reviewer

recommends this paper be published after major revisions.

The authors found a bug in the code that calculated the difference between V3 and the CERES data. As the global coverage varies with season (i.e no data in the polar winters) for the AATSR data, the data is **now** only compared with CERES when both instruments report data. The data has been reprocessed and the numbers in the table have been updated accordingly. The change to the numbers between -60 and 60 latitude was negligible however the change to the value encompassing -90 to 90 has changed considerably nearly all the comparisons with CERES data have improved . The text has been modified accordingly in the section 'Comparison of radiative fluxes'. All except the LW BOA down ( all sky and clearsky) agree within the CERES uncertainty estimates. The LW BOA estimates are of the order (2.8%  allsky and 3.8% clearly) just outside the range of the CERES uncertainty. It is hypothesised that the assumed cloud base height is systematically biased in the AATSR data set. This will be re-evaluated in future versions.

New figure below

[revised manuscript text omitted]

##############################################################################

---

## Author Comment (AC2) · 12 Jun 2020

The authors thank the reviewer for the comments which have improved the paper. The reviewer's comments are shown in red and the authors response shown in black.

In Figures 1 and 2, I recommend adding a third column which plots differences between

the versions.

A third column has been added to theses plots showing the differences, note that the plot style has changed slightly due to change in python plotting libraries.

New figure above, old figure below

[Figure]

**Figure 1.** Examples from 2008 of Level-3C (yearly average) Cloud_cci AATSR V3 (left), V2 (middle) and difference of V3-V2 (right). From the top: cloud fraction (CFC), liquid cloud faction (CPH), cloud optical thickness (COT) and cloud effective radius (CER),

**Figure 2.** As Fig. 1 but for cloud top height (CTH), liquid water path (LWP), ice water path (IWP) and cloud albedo (CLA).

[Figure]

**Figure 1.** Examples from 2008 of Level-3C (yearly average) Cloud_cci AATSR V3 (left) and V2 (right). From the top: cloud fraction (CFC), liquid cloud faction (CPH), cloud optical thickness (COT) and cloud effective radius (CER),

**Figure 2.** As Fig. 1 but for cloud top height (CTH), liquid water path (LWP), ice water path (IWP) and cloud albedo (CLA).

Table 3 contains a bunch of metrics that I don't know what they are. I think it would be

good to give a one-sentence definition of KSS, hit rate, and POD.

The text has been clarified with the additional text below

The Hanssen-Kuiper skill score (KSS), an often used skill score [Hansen 1965] is defined as KSS = TPR-FPR  where TPR is fraction of pixels correctly identified as cloud and FPR is the fraction of pixels wrongly identified as cloud. probability of detection (POD), the fraction of pixels identified correctly as clear from 61% to 76%.

Line 181 (and rest of paragraph): I don't know what "cost" is referring to, so "cost less

than 5" makes no sense. Line 188: they state here that thin multi-layer clouds are retrieved as the weighted

average of the two layers. It would be useful to say if this situation is flagged in the data

or if there's some mechanism to screen the data for this.

This section has ben clarified with the additional text below.

The cost is an out put of the optimal estimation retrieval scheme and is the result of the squared deviations between the measurements and the forward model (which in this scenario is a single layer of cloud) and the retrieved state vector and the a priori state vector, weighted by an associated covariance matrix. Essentially it is an indicator if the observed measurements were a good fit to the forward model. Cost less than 5 indicates the measurements fit the model well. A higher cost would indicate we are viewing cloud form multiple layers, for example.

From line 221 to the end of the section: they discuss hear the uncertainty of the radiative flux estimates. Does uncertainty here refer to precision or accuracy? Also, are the

values given for CERES (5.4 and 4.6 W/m2) the uncertainty for the global average or

at a grid point?

The authors found a bug in the code that calculated the difference between V3 and the CERES data. As the global coverage varies with season (i.e no data in the polar winters) for the AATSR data, the data is now only compared with CERES when both instruments report data. The data has been reprocessed and the numbers in the table have been updated accordingly. The change to the numbers between -60 and 60 latitude was negligible however the change to the value encompassing -90 to 90 has changed considerably nearly all the comparisons with CERES data have improved . The text has been modified accordingly in the section 'Comparison of radiative fluxes'. All except the LW BOA down ( all sky and clearsky) agree within the CERES uncertainty estimates. The LW BOA estimates are of the order (2.8%  allsky and 3.8% clearly) just outside the range of the CERES uncertainty. It is hypothesised that the assumed cloud base height is systematically biased in the AATSR data set. This will be re-evaluated in future versions.

The revised plots are shown below and referenced to the previous plots

New figure below

[Figure]

**Figure 4.** Examples of Level-3C (yearly average for 2008) Cloud_cci AATSR V3 (left column), CERES (middle column) and difference CERES-AATSR (right column) global maps of fluxes from top to bottom $\mathrm{LWF_{TOA}^{up}}$, $\mathrm{LWF_{TOA}^{up}clear}$, $\mathrm{SWF_{TOA}^{up}}$ and $\mathrm{SWF_{TOA}^{up}clear}$

**Figure 5.** Examples of Level-3C (yearly average for 2008) Cloud_cci AATSR V3 (left column) and CERES (middle column) and difference CERES-AATSR in the right column global maps of forcing from the top to the bottom, $\mathrm{LWF_{BOA}^{down}}$, $\mathrm{LWF_{BOA}^{down}clear}$, $\mathrm{SWF_{BOA}^{down}}$, $\mathrm{SWF_{BOA}^{down}clear}$

Old figures below

[Figure]

**Figure 4.** Examples of Level-3C (yearly average for 2008) Cloud_cci AATSR V3 (left column), CERES (middle column) and difference CERES-AATSR (right column) global maps of forcings from top to bottom $\mathrm{LWF_{TOA}^{up}}$, $\mathrm{LWF_{TOA}^{up}clear}$, $\mathrm{SWF_{TOA}^{up}}$ and $\mathrm{SWF_{TOA}^{up}clear}$

**Figure 5.** Examples of Level-3C (yearly average for 2008) Cloud_cci AATSR V3 (left column) and CERES (middle column) and difference CERES-AATSR in the right column global maps of forcing from the top to the bottom, $\mathrm{LWF_{BOA}^{down}}$, $\mathrm{LWF_{BOA}^{down}clear}$, $\mathrm{SWF_{BOA}^{down}}$, $\mathrm{SWF_{BOA}^{down}clear}$

New tables shown here

**Table 6.** Multi-annual (2003-2012), zonal averaged broadband shortwave and longwave fluxes (SWF, LWF) at the top-of-atmosphere (TOA) inferred from the Cloud_cci AATSR V3 dataset. Two latitude ranges, -60° to 60° (top) and -90° to 90° (bottom), are presented. The values are compared with the equivalent values from the Clouds and Earth Radiation Energy System (CERES) Energy Balanced and Filled (EBAF) fluxes. All values are given in $\mathrm{Wm}^{-2}$. The differences and relative differences are also reported.

| TOA flux comparison with CERES | | | | |
|---|---|---|---|---|
| | $\mathrm{LWF}^{up}_{TOA}$ | $\mathrm{SWF}^{up}_{TOA}$ | Clear $\mathrm{LWF}^{up}_{TOA}$ | Clear $\mathrm{SWF}^{up}_{TOA}$ |
| Cloud_cci ATSR-2/AATSRv3 | 245.8 | 104.4 | 268.7 | 47.5 |
| CERES EBAF Ed 4.1 | 244.1 | 98.70 | 273.9 | 48.8 |
| Difference | -1.7 | -5.7 | 5.2 | 1.3 |
| Rel. difference | 0.7% | 5.7% | 1.9% | 2.7% |
| Cloud_cci ATSR-2/AATSRv3 | 235.7 | 113.7 | 235.7 | 61.7 |
| CERES EBAF Ed 4.1 | 233.4 | 108.8 | 233.4 | 63.3 |
| Difference | -2.3 | -4.9 | -2.3 | 1.6 |
| Rel. difference | 1.0% | 4.5% | 1% | 2.5% |

**Table 7.** As for Table. 6 but for the bottom-of-atmosphere (BOA).

| BOA flux comparison with CERES | | | | |
|---|---|---|---|---|
| | $\mathrm{LWF}^{down}_{BOA}$ | $\mathrm{SWF}^{down}_{BOA}$ | $\mathrm{clearLWF}^{down}_{BOA}$ | $\mathrm{clearSWF}^{down}_{BOA}$ |
| Cloud_cci ATSR-2/AATSRv3 | 364.5 | 191.8 | 335.7 | 255.5 |
| CERES EBAF Ed 4.1 | 354.4 | 190.0 | 323.9 | 250.4 |
| Difference | -10.1 | 1.8 | -11.2 | -5.1 |
| Rel. Difference | 2.9% | .9% | 3.5% | 2.0% |
| Cloud_cci ATSR-2/AATSRv3 | 335.7 | 180.2 | 303.2 | 240.7 |
| CERES EBAF Ed 4.1 | 326.5 | 179.0 | 292.2 | 237.6 |
| Difference | -9.2 | -1.2 | -11.0 | -3.1 |
| Rel. Difference | 2.7% | .7% | 3.8% | 1.3% |

**Old tables for reference**

**Table 6.** Multi-annual (2003-2012), zonal averaged broadband shortwave and longwave fluxes (SWF, LWF) at the top-of-atmosphere (TOA) inferred from the Cloud_cci AATSR V3 dataset. Two latitude ranges, -60° to 60° (top) and -90° to 90° (bottom), are presented. The values are compared with the equivalent values from the Clouds and Earth Radiation Energy System (CERES) Energy Balanced and Filled (EBAF) fluxes. All values are given in $\mathrm{Wm}^{-2}$. The differences and relative differences are also reported.

| TOA flux comparison with CERES | | | | |
|---|---|---|---|---|
| | $\mathrm{LWF}^{up}_{TOA}$ | $\mathrm{SWF}^{up}_{TOA}$ | Clear $\mathrm{LWF}^{up}_{TOA}$ | Clear $\mathrm{SWF}^{up}_{TOA}$ |
| Cloud_cci ATSR-2/AATSRv3 | 246.3 | 104.4 | 268.7 | 61.9 |
| CERES EBAF Ed 4.1 | 244.2 | 98.70 | 273.9 | 63.6 |
| Difference | -1.9 | -5.7 | 5.2 | 1.63 |
| Rel. difference | 0.8% | 5.7% | 1.9% | 2.6% |
| Cloud_cci ATSR-2/AATSRv3 | 234.9 | 114.0 | 255.1 | 47.5 |
| CERES EBAF Ed 4.1 | 225.1 | 104.2 | 248.9 | 48.7 |
| Difference | -9.9 | -9.8 | -6.2 | 1.2 |
| Rel. difference | 4.4% | 9.4% | 2.4% | 2.7% |

**Table 7.** As for Table. 6 but for the bottom-of-atmosphere (BOA).

| BOA flux comparison with CERES | | | | |
|---|---|---|---|---|
| | $\text{LWF}_{\text{BOA}}^{\text{down}}$ | $\text{SWF}_{\text{BOA}}^{\text{down}}$ | $\text{clearLWF}_{\text{BOA}}^{\text{down}}$ | $\text{clearSWF}_{\text{BOA}}^{\text{down}}$ |
| Cloud_cci ATSR-2/AATSRv3 | 364.6 | 192.2 | 335.3 | 255.7 |
| CERES EBAF Ed 4.1 | 354.4 | 190.4 | 323.9 | 250.3 |
| Difference | -10.26 | 1.9 | -11.4 | -5.4 |
| Rel. Difference | 2.9% | .97% | 3.5% | 2.1% |
| Cloud_cci ATSR-2/AATSRv3 | 334.1 | 181.3 | 301.7 | 241.3 |
| CERES EBAF Ed 4.1 | 306.8 | 176.0 | 272.7 | 232.6 |
| Difference | -28.4 | -5.3 | -29.1 | -8.7 |
| Rel. Difference | 9.2% | 3.0% | 10% | 3.8% |

Captions of Fig. 4 and 5. The captions refer to "forcing", but they really mean are fluxes. This should be changed

This has been fixed

| | $\text{LWF}_{\text{BOA}}^{\text{down}}$ | $\text{SWF}_{\text{BOA}}^{\text{down}}$ | $\text{clearLWF}_{\text{BOA}}^{\text{down}}$ | $\text{clearSWF}_{\text{BOA}}^{\text{down}}$ |
|---|---|---|---|---|
| Cloud_cci ATSR-2/AATSRv3 | 334.1 | 181.3 | 301.7 | 241.3 |
| CERES EBAF Ed 4.1 | 306.8 | 176.0 | 272.7 | 232.6 |
| Difference | -28.4 | -5.3 | -29.1 | -8.7 |
| Rel. Difference | 9.2% | 3.0% | 10% | 3.8% |

---

## Author Comment (AC3) · 12 Jun 2020

The authors thank the reviewer for the comments which have undoubtedly improved the paper. The reviewer's comments are shown in red and the authors response shown in black.

One paper on solar calibration accuracy is referenced but references on calibration of the

infrared radiances are necessary, particularly in light of the statement that the dataset

is well suited to investigation of trends.

Added in reference for calibration of Infra red channels

D.L. Smith, J. Delderfield, D. Drummond, T. Edwards, C.T. Mutlow, P.D. Read, G.M. Toplis,

Calibration of the AATSR instrument, Advances in Space Research, Volume 28, Issue 1,

2001, Pages 31-39, ISSN 0273-1177, https://doi.org/10.1016/S0273-1177(01)00273-3.

Specific comments: The sentence in lines 45-47 implies CloudSat detection of GHGforced trends in cloud height is limited by nadir-only sampling. The larger issue is

separating forced cloud changes from natural variability. There has been considerable

work which makes clear that long-term calibration stability is a major difficulty in characterizing forced trends from passive sensors observations. See, for example, Shea et

al. (JGR, 2017, doi:10.1175/JCLI-D-16-0429.1). I'm not sure what the point of the last

half of this paragraph is, which raises complex observational issues which are beyond

the scope of this paper.

In response to the above comment I have removed this sentence and associated reference

'and they have operated for a relatively brief period. It has been estimated that CloudSat-like radar instruments would need to constantly observe the Earth until at least 2030 to detect a noticeable trend in cloud top height related to climate change~\citep{takahashi}.'

Line 72: This doesn't look like a complete sentence. Maybe something like "ATSR is

designed to provide low noise radiance measurements . . ." - ?

This paragraph is rewritten to be more clear

ATSR channels are specifically is designed to have low noise. Furthermore AATSR measurements are carried out with a high level of accuracy as the instrument includes an on-board thermal black body and a visible calibration system designed for high uniformity and stability~\citep{smitha}.

Table 3: Define 'hit rate'. How is hit rate different from probability of detection? I'm not familiar with Kuiper Skill Score, please provide a reference.

Definition of the hitrate was added to the text

hit rate the percentage of pixels identified correctly as either cloudy or clear

A reference for the Kuiper Skill score was added

Hanssen, A.W.; Kuipers, W.J.A. On the relationship between frequency of rain and various meteorological parameters. Meded. Verh. 1965, 81, 2–15.

Line 221: As a general comment: peer reviewed publications should be cited rather than data quality summaries posted on-line, unless the information is only available from on-line summaries. Results reported in refereed papers are archival, have usually received more scrutiny, and tend to be better explained and documented.

Updated the reference for TOA CERES  uncertainty to Loeb 2018 however a published reference for the uncertainty of BOA products could  not be identified so the data quality statement (with link now referenced) has been included.

Text changed:

The CERES team have evaluated the accuracy of the products in [Loeb et al 2018] and states that their all-sky shortwave and longwave **monthly** uncertainty is 2.5(3)  Wm-2 for Aqua and Terra (Terra only) period, while the clear-sky shortwave and longwave uncertainty is 5(6) and 4.5(5) Wm-2, respectively for the Aqua and Terra (Terra only) monthly products

Line 222: Are the CERES uncertainties which are mentioned the uncertainties in the monthly global means?

Answered in the paragraph added above, they are monthly uncertainties

Line 223 and 258-259: global means are within CERES uncertainties only for 60N60S. All-sky fluxes show differences which are much larger, and there are significant regional biases which seem to be associated with clouds.

 The authors found a bug in the code that calculated the difference between V3 and the CERES data. As the global coverage varies with season (i.e no data in the polar winters) for the AATSR data, the data is now only compared with CERES when both instruments report data. The data has been

reprocessed and the numbers in the table have been updated accordingly. The change to the numbers between -60 and 60 latitude was negligible however the change to the value encompassing -90 to 90 has changed considerably nearly all the comparisons with CERES data have improved . The text has been modified accordingly in the section 'Comparison of radiative fluxes'. All except the LW BOA down ( all sky and clearsky) agree within the CERES uncertainty estimates. The LW BOA estimates are of the order (2.8%  allsky and 3.8% clearly) just outside the range of the CERES uncertainty. It is hypothesised that the assumed cloud base height is systematically biased in the AATSR data set. This will be re-evaluated in future versions.

New figure below

[Figure]

**Figure 4.** Examples of Level-3C (yearly average for 2008) Cloud_cci AATSR V3 (left column), CERES (middle column) and difference CERES-AATSR (right column) global maps of fluxes from top to bottom $LWF_{TOA}^{up}$, $LWF_{TOA}^{up}$clear, $SWF_{TOA}^{up}$ and $SWF_{TOA}^{up}$clear

**Figure 5.** Examples of Level-3C (yearly average for 2008) Cloud_cci AATSR V3 (left column) and CERES (middle column) and difference CERES-AATSR in the right column global maps of forcing from the top to the bottom, $LWF_{BOA}^{down}$, $LWF_{BOA}^{down}$clear, $SWF_{BOA}^{down}$, $SWF_{BOA}^{down}$clear

Old figures below

[Figure]

**Figure 4.** Examples of Level-3C (yearly average for 2008) Cloud_cci AATSR V3 (left column), CERES (middle column) and difference CERES-AATSR (right column) global maps of forcings from top to bottom $LWF_{TOA}^{up}$, $LWF_{TOA}^{up}$clear, $SWF_{TOA}^{up}$ and $SWF_{TOA}^{up}$clear

**Figure 5.** Examples of Level-3C (yearly average for 2008) Cloud_cci AATSR V3 (left column) and CERES (middle column) and difference CERES-AATSR in the right column global maps of forcing from the top to the bottom, $LWF_{BOA}^{down}$, $LWF_{BOA}^{down}$clear, $SWF_{BOA}^{down}$, $SWF_{BOA}^{down}$clear

New tables shown here

**New tables shown here**

**Table 6.** Multi-annual (2003-2012), zonal averaged broadband shortwave and longwave fluxes (SWF, LWF) at the top-of-atmosphere (TOA) inferred from the Cloud_cci AATSR V3 dataset. Two latitude ranges, $-60°$ to $60°$ (top) and $-90°$ to $90°$ (bottom), are presented. The values are compared with the equivalent values from the Clouds and Earth Radiation Energy System (CERES) Energy Balanced and Filled (EBAF) fluxes. All values are given in $\mathrm{Wm^{-2}}$. The differences and relative differences are also reported.

| TOA flux comparison with CERES | | | | |
|---|---|---|---|---|
| | $\mathrm{LWF^{up}_{TOA}}$ | $\mathrm{SWF^{up}_{TOA}}$ | Clear $\mathrm{LWF^{up}_{TOA}}$ | Clear $\mathrm{SWF^{up}_{TOA}}$ |
| Cloud_cci ATSR-2/AATSRv3 | 245.8 | 104.4 | 268.7 | 47.5 |
| CERES EBAF Ed 4.1 | 244.1 | 98.70 | 273.9 | 48.8 |
| Difference | -1.7 | -5.7 | 5.2 | 1.3 |
| Rel. difference | 0.7% | 5.7% | 1.9% | 2.7% |
| Cloud_cci ATSR-2/AATSRv3 | 235.7 | 113.7 | 235.7 | 61.7 |
| CERES EBAF Ed 4.1 | 233.4 | 108.8 | 233.4 | 63.3 |
| Difference | -2.3 | -4.9 | -2.3 | 1.6 |
| Rel. difference | 1.0% | 4.5% | 1% | 2.5% |

**Table 7.** As for Table. 6 but for the bottom-of-atmosphere (BOA).

| BOA flux comparison with CERES | | | | |
|---|---|---|---|---|
| | $\mathrm{LWF^{down}_{BOA}}$ | $\mathrm{SWF^{down}_{BOA}}$ | $\mathrm{clearLWF^{down}_{BOA}}$ | $\mathrm{clearSWF^{down}_{BOA}}$ |
| Cloud_cci ATSR-2/AATSRv3 | 364.5 | 191.8 | 335.7 | 255.5 |
| CERES EBAF Ed 4.1 | 354.4 | 190.0 | 323.9 | 250.4 |
| Difference | -10.1 | 1.8 | -11.2 | -5.1 |
| Rel. Difference | 2.9% | .9% | 3.5% | 2.0% |
| Cloud_cci ATSR-2/AATSRv3 | 335.7 | 180.2 | 303.2 | 240.7 |
| CERES EBAF Ed 4.1 | 326.5 | 179.0 | 292.2 | 237.6 |
| Difference | -9.2 | -1.2 | -11.0 | -3.1 |
| Rel. Difference | 2.7% | .7% | 3.8% | 1.3% |

**Old tables for reference**

**Table 6.** Multi-annual (2003-2012), zonal averaged broadband shortwave and longwave fluxes (SWF, LWF) at the top-of-atmosphere (TOA) inferred from the Cloud_cci AATSR V3 dataset. Two latitude ranges, $-60°$ to $60°$ (top) and $-90°$ to $90°$ (bottom), are presented. The values are compared with the equivalent values from the Clouds and Earth Radiation Energy System (CERES) Energy Balanced and Filled (EBAF) fluxes. All values are given in $\mathrm{Wm^{-2}}$. The differences and relative differences are also reported.

| TOA flux comparison with CERES | | | | |
|---|---|---|---|---|
| | $\mathrm{LWF^{up}_{TOA}}$ | $\mathrm{SWF^{up}_{TOA}}$ | Clear $\mathrm{LWF^{up}_{TOA}}$ | Clear $\mathrm{SWF^{up}_{TOA}}$ |
| Cloud_cci ATSR-2/AATSRv3 | 246.3 | 104.4 | 268.7 | 61.9 |
| CERES EBAF Ed 4.1 | 244.2 | 98.70 | 273.9 | 63.6 |
| Difference | -1.9 | -5.7 | 5.2 | 1.63 |
| Rel. difference | 0.8% | 5.7% | 1.9% | 2.6% |
| Cloud_cci ATSR-2/AATSRv3 | 234.9 | 114.0 | 255.1 | 47.5 |
| CERES EBAF Ed 4.1 | 225.1 | 104.2 | 248.9 | 48.7 |
| Difference | -9.9 | -9.8 | -6.2 | 1.2 |
| Rel. difference | 4.4% | 9.4% | 2.4% | 2.7% |

**Table 7.** As for Table. 6 but for the bottom-of-atmosphere (BOA).

| | $\text{LWF}_{\text{BOA}}^{\text{down}}$ | $\text{SWF}_{\text{BOA}}^{\text{down}}$ | $\text{clearLWF}_{\text{BOA}}^{\text{down}}$ | $\text{clearSWF}_{\text{BOA}}^{\text{down}}$ |
|---|---|---|---|---|
| | | BOA flux comparison with CERES | | |
| Cloud_cci ATSR-2/AATSRv3 | 364.6 | 192.2 | 335.3 | 255.7 |
| CERES EBAF Ed 4.1 | 354.4 | 190.4 | 323.9 | 250.3 |
| Difference | -10.26 | 1.9 | -11.4 | -5.4 |
| Rel. Difference | 2.9% | .97% | 3.5% | 2.1% |
| Cloud_cci ATSR-2/AATSRv3 | 334.1 | 181.3 | 301.7 | 241.3 |
| CERES EBAF Ed 4.1 | 306.8 | 176.0 | 272.7 | 232.6 |
| Difference | -28.4 | -5.3 | -29.1 | -8.7 |
| Rel. Difference | 9.2% | 3.0% | 10% | 3.8% |

Line 164: "version 4-20" should be "version 4.20"

Corrected in text

Line 228: incomplete sentence

Paragraph rephrased to the text below

The TOA shortwave flux in clear scenes is systematically lower than CERES indicating a potential underestimate of the surface reflectance in the AATSR product. The AATSR surface reflectance model uses a Cox and Munk~\cite{cox} formulation a key source of uncertainty could be the sensitivity to the diurnal correction applied to the AATSR data in order to make a like for like comparison with CERES. These differences his will be investigated for improvement in future versions.